# Oncometabolism: A Paradigm for the Metabolic Remodeling of the Failing Heart

**DOI:** 10.3390/ijms232213902

**Published:** 2022-11-11

**Authors:** Annika-Ricarda Kuhn, Marc van Bilsen

**Affiliations:** Department of Physiology, Cardiovascular Research Institute Maastricht (CARIM), Maastricht University, P.O. Box 616, 6200 MD Maastricht, The Netherlands

**Keywords:** cardiac hypertrophy, energy metabolism, glycolysis, Warburg effect, cell signaling

## Abstract

Heart failure is associated with profound alterations in cardiac intermediary metabolism. One of the prevailing hypotheses is that metabolic remodeling leads to a mismatch between cardiac energy (ATP) production and demand, thereby impairing cardiac function. However, even after decades of research, the relevance of metabolic remodeling in the pathogenesis of heart failure has remained elusive. Here we propose that cardiac metabolic remodeling should be looked upon from more perspectives than the mere production of ATP needed for cardiac contraction and relaxation. Recently, advances in cancer research have revealed that the metabolic rewiring of cancer cells, often coined as oncometabolism, directly impacts cellular phenotype and function. Accordingly, it is well feasible that the rewiring of cardiac cellular metabolism during the development of heart failure serves similar functions. In this review, we reflect on the influence of principal metabolic pathways on cellular phenotype as originally described in cancer cells and discuss their potential relevance for cardiac pathogenesis. We discuss current knowledge of metabolism-driven phenotypical alterations in the different cell types of the heart and evaluate their impact on cardiac pathogenesis and therapy.

## 1. Introduction

Cardiovascular disease remains a major cause of morbidity and mortality in both developing and developed countries worldwide. Heart failure (HF) afflicts 27 million patients globally and despite advances in treatment, it still constitutes a rising socioeconomic burden, primarily due to an expanding, aging population [1]. Current HF treatment is primarily targeted at the reduction of cardiac workload via the inhibition of neuroendocrine pathways. However, derangements in cardiac energy metabolism have also been implicated in the pathogenesis of HF, and there is an increasing interest in the modulation of cardiac metabolism as an alternative or complementary therapeutic approach [2].

The healthy heart predominantly relies on the mitochondrial oxidation of fatty acids (60–90%) and of glucose and lactate (10–40%) for the generation of adenosine 5′-triphosphate (ATP), which is necessary to sustain the contraction–relaxation cycle of the cardiac muscle [3]. The contribution of each of these substrates to cardiac ATP production varies and to a large extent depends on extra-cardiac factors (exercise, diet, fasting, etc.). This flexibility in substrate utilization allows the healthy heart to dynamically alter fatty acid and carbohydrate metabolism depending on substrate availability. Increases in circulating fatty acid levels decrease the uptake of glucose and vice versa [4]. During HF this metabolic flexibility becomes compromised. In the failing heart, the ability to oxidize fatty acids is reduced, while the glycolytic conversion of glucose to lactate increases [5]. Mechanistically, this shift away from mitochondrial oxidative metabolism, and consequently the reduction in ATP production, is commonly considered to compromise cardiac function. The failing heart is believed to become “energy-starved” [6,7], as reflected, among other things, by the decline in the cardiac creatine phosphate (CrP) and ATP levels, and in the CrP/ATP ratio in HF patients as assessed by NMR [8,9,10]. However, it is still debated whether the observed reduction in cardiac high-energy phosphate levels in HF patients is sufficient to cause serious cardiac dysfunction [11].

In this review, we investigate substrate and energy metabolism from an entirely different perspective and consider whether the rewiring of cardiac substrate metabolism may have consequences far beyond the mere synthesis of ATP needed to maintain adequate cardiac muscle contraction. Recent developments in cancer research have convincingly demonstrated that the rewiring of cellular substrate metabolism is a crucial event in driving cellular phenotype [12] and parallels have been drawn recently between oncometabolism and substrate metabolism of the failing heart [13]. Here, we discuss the implications of these novel insights regarding the rewiring of metabolic fluxes for cardiac disease. Thereto, we will discuss the principles of influential metabolic changes in the main metabolic pathways first described in cancer, such as the Warburg effect. Next, we review the evidence for metabolic control of cellular phenotype in the main cell types of the cardiovascular system, namely fibroblasts, endothelial cells, vascular smooth muscle cells, immune cells and cardiomyocytes. Finally, the potential implications of the metabolic rewiring for the pathogenesis and treatment of cardiac disease are discussed.

## 2. Metabolic Rewiring in Cancer: Possible Implications for the Heart

Most of our current knowledge regarding the influence of metabolic changes on cell phenotype and function stems from research on cancer cells. These groundbreaking studies revealed that reprogramming metabolic pathways ensures the survival and proliferation of malignant cells. Non-dividing differentiated cells largely rely on mitochondrial oxidative phosphorylation to maintain cellular homeostasis. On the other hand, rapidly proliferating, undifferentiated cells, such as cancer cells, are characterized by high rates of glycolysis and increased metabolism of glutamine (glutaminolysis) [14]. The reprogramming of these metabolic pathways, often referred to as oncometabolism, therefore constitutes a hallmark of cancer and is essential for the growth of a tumor [15]. During neoplasia, cancer cells display high rates of metabolic turnover and switch from mainly catabolic to anabolic pathways in order to facilitate the generation of precursor molecules for rapid cell growth and proliferation. The most acknowledged change in cancer cell metabolism is metabolic switching to high rates of glycolysis and increased lactate production. In the cardiovascular field, high rates of glycolysis are traditionally associated with hypoxia or ischemia (anaerobic glycolysis), but in tumors, glycolysis can be upregulated when oxygen is abundantly present. This is commonly referred to as aerobic glycolysis. Aerobic glycolysis was first documented in tumor cells by Otto Warburg almost a century ago and is therefore referred to as the Warburg effect [16]. The question that emerges is if the heart also has the intrinsic capacity to increase glycolysis under conditions when oxygen availability is not limiting. In other words, is the Warburg effect operational in the cardiac muscle?

The shift from mitochondrial oxidative to glycolytic metabolism observed in tumors is reminiscent of the shift from fatty acid to glucose metabolism observed in the hypertrophic and failing heart [11,17,18]. In cancer cells, enhanced glycolysis supports an anabolic, proliferative state of the cells, even if it is at the expense of total ATP synthesis. It has been estimated that the normal beating heart invests 20–30% of its ATP in basal metabolism and in maintaining cellular integrity [19]. It is important to realize that cardiac remodeling also depends on growth (cardiomyocyte hypertrophy), proliferation of fibroblasts (fibrosis) and endothelial cells (angiogenesis), as well as infiltration of immune cells (migration). Furthermore, it has also been demonstrated that cardiomyocyte hypertrophy is associated with the partial re-activation of the cell cycle and this is often considered an attempt to initiate proliferation [20].

This raises a number of intriguing questions. Is the metabolic remodeling observed in the hypertrophic and failing heart meant to optimize energy metabolism (and thus pump function), or does it rather serve to support anabolic processes and to allow cardiac cells to adapt their phenotype? Which cell types constituting the heart are subject to metabolic rewiring? What implications does this all have in the setting of insulin resistance and diabetes, when the cardiac muscle becomes even more dependent on the oxidation of fatty acids? To address these questions, we first discuss the various metabolic pathways that contribute to the metabolic rewiring in cancer cells in more detail.

### 2.1. The Warburg Effect

In terms of energy yield, the conversion of glucose to lactate is far less efficient than the complete oxidation of glucose, yielding only 2 ATP instead of up to 36 ATP [21]. Despite the lower ATP yield, the switch toward aerobic glycolysis is considered highly advantageous for tumor cells. An increased glucose uptake promotes increased flux through pathways that use glycolytic intermediates as a starting point. This includes the pentose phosphate pathway (PPP) and the hexosamine biosynthetic pathway (HBP) that use glucose-6-phosphate and fructose-6-phosphate as substrates (Figure 1). The PPP leads to the formation of precursor molecules of nucleic acids, needed by proliferating cells. A substantial part of the glycolytic intermediates is also used for the synthesis of amino acids, thereby stimulating protein synthesis [22].

Originally, the Warburg effect was attributed to mitochondrial dysfunction. Later studies showed that mitochondrial oxidative capacity in cancer cells was marginally affected in many cases [16], suggesting that direct activation of the glycolytic pathway was the primary cause of the Warburg effect. Although various signaling pathways (PI3K-Akt-mTOR) and transcription factors such as Forkhead, the tumor suppressor p53 and the oncogene Myc have been implicated in the stimulation of aerobic glycolysis [23,24], the transcription factor hypoxia-inducible factor-1a (HIF-1a) is considered the major driver of the Warburg effect.

The activity of HIF-1a is primarily regulated at the level of protein stability and, as the name implies, is dependent on oxygen levels. HIF-1a stability is controlled by the prolyl hydroxylase (PHD)-dependent hydroxylation of proline residues in HIF-1a. This reaction requires oxygen and alpha-ketoglutarate as substrates and yields succinate as an end product (Figure 2). Under normoxic conditions, the PHD-dependent hydroxylation of HIF-1a marks it for ubiquitin-mediated degradation. However, when oxygen is limiting, PHD activity is suppressed, resulting in HIF-1a stabilization and subsequent induction of HIF-1a-dependent genes [25]. Notably, the end product succinate, which also is an intermediate of the tricarboxylic acid (TCA) cycle, acts as an inhibitor of PDH activity and thereby causes stabilization of HIF-1a. This implies that independent of oxygen levels, a rise in cellular succinate levels—for instance, due to changes in TCA cycle activity—also prevents HIF-1a degradation. The latter provides a mechanistic explanation for the Warburg effect. Hypoxia-independent stabilization of HIF-1a by changes in cellular succinate levels was first documented in cancer cells [26,27,28] and later shown to be operative in macrophages as well [29]. These studies reveal the close interconnection between TCA cycle activity and HIF stability.

HIF-1a stabilization results in the concerted induction of the glucose transporter (GLUT1), nearly all glycolytic enzymes (Table 1) and the monocarboxylate transporter-1 (MCT-1) involved in the export of lactate. It also induces the expression of the kinase pyruvate dehydrogenase kinase-1 (PDK1). PDK1 phosphorylates the pyruvate dehydrogenase (PDH) complex, thereby inhibiting PDH and diverting pyruvate away from mitochondrial oxidation [30]. Finally, it induces the expression of angiogenic factors like angiopoietin-like-4 (ANGPTL4) and vascular endothelial growth factor (VEGF) [31,32]. ANGPTL4 also inhibits the activity of lipoprotein lipase on the endothelial luminal surface and, in this way, limits the delivery of fatty acids as a substrate for mitochondrial oxidation [33].

Accordingly, HIF-1a promotes the use of glucose at all levels, from the import of glucose down to the export of lactate. Moreover, at the same time, it inhibits the mitochondrial oxidation of glucose (via PDK1) and indirectly that of fatty acids (via ANGPTL4), and promotes oxygen supply to cells (via VEFG-stimulated angiogenesis).

It is well feasible that similar mechanisms are operative in the myocardium. Perhaps the commonly observed increase in glycolysis and its partial uncoupling from glucose oxidation during cardiac hypertrophy and failure [17,34,35] are indications of the Warburg effect being operative in the diseased heart.

### 2.2. Non-Canonical Functions of Glycolytic Enzymes

When discussing the glycolytic pathway from a metabolic perspective, one also has to take into account that many of the glycolytic enzymes have been shown to have pleiotropic functions and are directly involved in cellular signaling. These reported actions are listed in Table 1. For instance, the expression of hexokinase-2 (HK2), the isozyme predominantly found in insulin-sensitive tissues like the heart, markedly increases during tumorigenesis [36,37]. Activation of the Akt-mTOR pathway evokes a HIF-1a-dependent increase in the expression of HK2 [38,39]. The isozyme is found both in the cytoplasm and bound to mitochondria, depending on cellular conditions.

**Table 1 ijms-23-13902-t001:** Non-canonical functions of glycolytic enzymes.

Enzyme Isoforms	TF Regulation	Non-Canonical Functions	Ref.
HK-1/2	HIF/Myc	Apoptosis	[40,41,42]
GPI	HIF	Secreted growth factor (cell migration), Apoptosis	[43,44,45]
PFK-M/L/P	HIF/Myc	YAP/TAZ Signaling	[45,46,47,48]
ALDO-A/C	HIF	Activates AMPK	[49,50,51]
TPI	HIF	n.d.	[52]
GAPDH	HIF	RNA-binding, Apoptosis	[53,54]
PGK-1	HIF	Protein kinase activity (autophagy)	[55,56]
PGAM-1	HIF	Actin binding (cell migration)	[57]
ENO-1/2	HIF/Myc	Apoptosis, cell migration, tRNA transport	[58,59,60]
PKM-1/2	HIF/Myc	Protein kinase activity	[61,62]

Importance of the transcription factors (TFs) HIF-1a and Myc in the transcriptional regulation of isoforms of glycolytic enzymes and reported non-canonical functions of these enzymes. For TPI, non-canonical functions have not been described (n.d.). HK—hexokinase, GPI—phosphoglucose isomerase, PFK—phosphofructokinase, ALDO—aldolase, TPI—triose-phosphate isomerase, GAPDH—glyceraldehyde-3-phosphate dehydrogenase, PGK—phosphoglycerate kinase, PGAM -phosphoglycerate mutase, ENO—enolase, PKM—pyruvate kinase (muscle isoform). Numbers and capitals refer to the different isoforms of these enzymes.

Following its phosphorylation by Akt, HK2 translocates to the outer mitochondrial membrane [63]. This inhibits apoptosis by interfering with the mitochondrial calcium- and reactive oxygen species (ROS)-mediated opening of the mitochondrial permeability transition pore [40,41]. Indeed, overexpression of HK2 promoted glycolysis and suppressed cancer cell death, an effect that depended on its N-terminal mitochondrial binding domain [64,65]. The enolase-1 isozyme (ENO1) is another glycolytic enzyme that is increased in metastatic cancer cells. Remarkably, ENO1 is also found on the cell surface of these cells where it interacts with plasminogen, thereby promoting ECM degradation and cell migration [58]. It would be interesting to explore if ENO1 has comparable non-canonical functions in cardiac fibrosis.

The pyruvate kinase muscle isozyme (PKM) catalyzes the last step of the glycolytic pathway. Alternative mRNA splicing of PKM gives rise to two mutually exclusive isoforms, containing either exon 9 (PKM1) or exon 10 (PKM2). PKM1 is normally found in more differentiated cells, whereas PKM2 is considered the embryonic form and is present at higher levels in proliferating cells, including cancer cells [66,67]. Both PKM1 and PKM2 form tetramers with high enzymatic activity. However, PKM2 is also present in the enzymatically less active dimeric and monomeric form. Not only does it divert glucose metabolism toward anabolic pathways, but it also exerts many other cellular functions as well. PKM2 is a target of tyrosine kinases [68] and interacts with phosphorylated tyrosine residues of proteins [69]. Both events interfere with the binding of the allosteric activator fructose-1,6-bisphosphate to PKM and thereby prevent the formation of the active tetrameric form. Through this, PKM activity and function are directly linked to growth factor-dependent tyrosine kinase signaling. Monomeric PKM2 has been shown to translocate toward the nucleus upon ERK1/2 phosphorylation, where it modulates gene transcription by acting as a co-activator of HIF-1a and of the beta-catenin/TCF4 complex [70,71]. Beta-catenin induces transcription of oncogenic Myc, a transcriptional regulator of several glycolytic enzymes (Table 1), and of Cyclin-D1, required for cell cycle progression. Via this mechanism, the formation of PKM2 by alternative splicing can be considered a hallmark of the Warburg effect as it promotes glycolysis and the use of glycolytic intermediates for biosynthetic purposes [72].

Notably, the development of cardiac hypertrophy is associated with the reactivation of the fetal gene program. This includes a general increase in the expression of various glycolytic enzymes, along with an isoform shift of several glycolytic enzymes to their fetal isoforms, including enolase (ENO3→ENO1) [73,74] and PKM (PKM1→PKM2) [75,76]. The functional significance of these isoform changes for the heart remains to be established. As far as HK2 is concerned, the non-canonical function of this enzyme in protecting the heart against apoptosis has been firmly established [77,78]. Strikingly, selective pharmacological activation of the PKM2 isoform protected against diabetes-induced kidney dysfunction [79]. It remains to be tested if PKM2 activation has similar effects in heart disease. Collectively, these studies suggest that the switch of glycolytic enzymes from one isoform to another not only has effects on glycolytic rate control but also serves discrete roles in cellular signaling.

### 2.3. Pentose Phosphate Pathway

As discussed, increased uptake and conversion of glucose to glucose-6-phosphate also fuels the PPP. The PPP promotes cancer cell proliferation by using glucose-6-phosphate and fructose-6-phosphate to generate ribose-5-phosphate via the oxidative branch (oxPPP) and the nonoxidative branch (non-oxPPP) of the PPP, respectively (Figure 1) [80]. Ribose-5-phosphate provides the substrate for the de novo synthesis of nucleotides that become incorporated in DNA, RNA, NAD(P) and ATP. The PPP gets activated in case of an increased requirement for nucleotide synthesis in highly proliferative cancerous cells. In addition, in the oxidative branch of the PPP, NADP is reduced to NADPH. NADPH is required for the de novo synthesis of fatty acids. The fatty acids, in turn, act as building blocks for phospholipids, needed for the formation of cellular membranes by proliferating cells. Furthermore, NADPH provides protection against oxidative stress by converting the antioxidant glutathione (GSH) to its reduced form [81].

In the heart, modifications in oxPPP flux have been implicated in the maintenance of the redox balance: Following pressure overload as well as myocardial infarction, mice exhibited increased activity of glucose-6-phosphate-dehydrogenase (G6PDH), the first and rate-limiting enzyme of the oxPPP, indicative of an increased oxPPP flux. G6PDH-deficient mice, in which oxPPP flux is reduced, exhibited exacerbated cardiac remodeling in response to myocardial infarction and pressure overload and experienced increased redox stress [82]. Rats that developed left ventricular hypertrophy and failure exhibited enhanced PPP flux as well, as evidenced by the accumulation of PPP intermediates and increased G6PDH activity [83]. Additionally, overexpression of hexokinase in mice amplified PPP flux and attenuated cardiac hypertrophy in response to isoproterenol [84]. On the other hand, in a dog model of pacing-induced heart failure, it was observed that the increase in G6PDH supplies the NADPH needed by ROS-generating enzymes and thereby fuels oxidative stress [85].

### 2.4. Hexosamine Biosynthetic Pathway

The HBP utilizes the glycolytic intermediate fructose-6-phosphate as substrate and accounts for 2–5% of total glucose metabolism. Enhanced flux through the HBP has been documented in cancer cells [86,87,88]. The HBP is highly dependent on glutamine availability, as glutamine is converted together with fructose-6-phosphate into glucosamine-6-phosphate(GlcN-6P) and glutamate by the rate-limiting enzyme glutamine:fructose-6-phosphate aminotransferase (GFAT) (Figure 1) [89]. As discussed in more detail later, in this way the glycolytic pathway and HBP are linked to glutamine metabolism. The end product of the HBP is UDP-N-acetylglucosamine (UDP-GlcNAc). Importantly, UDP-GlcNAc is a substrate for post-translational modifications of proteins by N-glycosylation, O-glycosylation and O-GlcNAcylation, which are essential for the regulation of cellular functions. For example, modification and subsequent activity modulation of glycolytic and lipid biosynthesis-associated enzymes by O-GlcNAcs have been documented in various cancers as prominent modulators of cellular metabolic pathways [90,91]. Specifically, O-GlcNAcylation of the glycolytic enzyme phosphofructokinase (PFK-1), a key regulatory enzyme in the glycolytic pathway, inhibits PFK-1 activity and diverts glycolytic flux toward the PPP, proving advantageous for cancer growth under conditions of oxidative stress [92]. In addition, the O-GlcNAcylation of proteins at Ser/Thr residues may compete with phosphorylation at the same site and may have divergent effects. For instance, O-GlcNAcylation of the oncogene Myc at Thr58 prevents phosphorylation of Thr58 and blocks its ubiquination and degradation [93,94]. Taken together, via the O-GlcNAcylation of proteins the HBP connects cellular metabolism with signaling pathways that are critical for cell growth and proliferation.

As far as the heart is concerned, it has been shown that various proteins involved in cardiac contraction, including Troponin I and T, are subject to O-GlcNAcylation [95]. In addition, it has been documented that challenging the cardiomyocytes or the heart in situ with pro-hypertrophic stimuli increases global protein O-GlcNAcylation [96]. Global O-GlcNAcylation is increased in cardiac tissue of advanced heart failure patients, patients with aortic stenosis as well as in murine models of cardiac hypertrophy and failure [97,98]. This increase in O-GlcNAcylation is, at least in murine hearts, accompanied by elevated levels of the cardiac O-GlcNAc transferase [99]. Pharmacological inhibition or cardiac deletion of O-GlcNAc transferase exacerbated cardiac dysfunction in association with decreased global O-GlcNAcylation [97,98]. Conversely, pharmacologically preserved O-GlcNAc levels in isolated adult cardiomyocytes provided protection against H_2_O_2_-induced cell death [100]. Evidence is accumulating that the modulatory effect of important regulatory proteins such as AMPK and HDAC4 on cardiac hypertrophy may be mediated by their effect on the HBP [96,101]. Enhanced flux through the HBP and consequent chronic GFAT activation has been suggested to contribute to the development of cardiac hypertrophy, as GFAT-associated O-GlcNAcylation appears to induce mTOR-mediated pro-hypertrophic signaling [102].

### 2.5. Glutaminolysis

Next to glucose, glutamine is an important carbon source for the biosynthesis of macromolecules required for the proliferation and maintenance of redox capacities within cancer cells [103]. Glutamine is the most abundant non-essential amino acid in circulation and is the main source of nitrogen needed for nucleotide synthesis. Glutamine fuels glutaminolysis, which is one of the major anaplerotic pathways. During glutaminolysis, glutamine is first converted into glutamate and ammonia via the enzyme glutaminase (Figure 3). Glutamate is then further converted into the TCA intermediate alpha-ketoglutarate by glutamate dehydrogenase, thereby fueling the TCA cycle and producing carbon sources for the generation of macromolecules (Figure 3) [104]. Reductive carboxylation of alpha-ketoglutarate yields citrate that acts as a source of cytosolic acetyl-CoA needed for the biogenesis of fatty acids and phospholipids [105]. Reductive carboxylation in tumor cells is believed to become important under conditions when HIF-1a is active or when mitochondrial function is impaired [106].

The role of glutaminolysis in the heart has not been clearly defined. Under normal conditions, the role of glutamine in the anaplerosis of the TCA cycle intermediates seems limited [107]. However, cardiac glutaminolysis was shown to be activated in a rat model of right ventricular hypertrophy [108]. Notably, the induction of transporters and enzymes involved in glutamine handling is Myc-dependent, again pointing to an important role of this oncogene in metabolic rewiring [109].

### 2.6. Metabolism and Post-Translational Modifications

In addition to UDP-GlcNAc, an end-product of the HBP pathway discussed above, acetyl-CoA also serves as an important metabolic intermediate for the post-translational modification of proteins. Protein acetylation occurs in the mitochondrial, cytoplasmic as well as nuclear compartments. In the cytosol, mitochondria-derived citrate is converted into acetyl-CoA and oxaloacetate by the enzyme ATP-citrate lyase (ACLY) (Figure 3) [110]. The acetyl unit can be used for lipogenesis or for the acetylation of lysine-residues in proteins via lysine acetyltransferases [110]. Acetylation affects not only the activity and stability of a large number of metabolic enzymes but also that of important regulatory proteins, such as the tumor suppressor p53 [111] and the peroxisome proliferator-activated receptor gamma, coactivator PGC-1a [112], the latter acting as a master regulator of mitochondrial biogenesis and respiration. In addition, the acetylation/deacetylation of histones provides epigenetic control of gene transcription [113].

Interestingly, the spectrum of posttranslational mechanisms associated with metabolic intermediates was recently extended by the discovery of another and comparable posttranslational modification process, which is based on the addition of succinate to lysine residues [114]. The substrate for this process appears to be the TCA cycle intermediate succinyl-CoA, but this still remains to be proven definitively. Whereas acetylation neutralizes the positive charge of lysine residues, succinylation converts it into a negatively charged residue. Interestingly, increasing the degree of protein succinylation, via genetic deletion of the desuccinylating enzyme Sirtuin-5 in mice, has been associated with the development of hypertrophic cardiomyopathy [115], suggesting that succinylation might also be of importance for the heart.

The combined data demonstrate that in cancer cells, alterations in substrate and energy metabolism directly influence the behavior of these cells. On the one hand, these alterations affect energy supply and balance, on the other, they modulate cellular phenotype and thereby cellular function. Changes in the fluxes through metabolic pathways provide substrates for the synthesis of biomolecules and for the post-translational modification of proteins. Metabolic enzymes themselves can also exert non-canonical functions and thereby regulate non-metabolic cellular processes.

## 3. Metabolic Rewiring in Cells of the Diseased Heart

Could it be that mechanisms similar to the metabolic rewiring in cancer cells are in place in cells of the cardiovascular system and play a role in cardiac disease? In the following section, we will reflect on the regulation of cellular phenotypes by metabolic rewiring in the principal cell types of the cardiac tissue, namely endothelial cells, fibroblasts, vascular smooth muscle cells and the cardiomyocytes themselves. We will first discuss the role of infiltrating immune cells, macrophages in particular, not only because these cells play an important role in cardiac disease, but equally relevant, because the concept of metabolic rewiring as a determinant of cell function has been developed further for immune cells. In fact, it led to the development of a new niche in immunology research: immunometabolism [116].

### 3.1. Immune Cell Metabolism

It is commonly accepted that the rewiring of immune cell metabolism is an integral and crucial element of the immune response [117]. Indeed, the fate and function of various types of immune cells, including macrophages, dendritic cells and T-cells, is determined by changes in cellular metabolism.

The connection between metabolic cues and macrophage phenotype and function has been a focal point of interest in recent years [118,119]. Macrophages adjust their phenotype in response to environmental stimuli, such as tissue injury, microbial infection, or T-cell signaling [120]. Historically, based on in vitro experiments, a distinction is made between two polarized states of the macrophages. Classically activated M1 macrophages exert pro-inflammatory functions to protect against microbial infections, whereas alternatively activated M2 macrophages serve a more anti-inflammatory role and coordinate tissue repair [121]. Recently, it has been recognized that macrophage polarization in vivo does not confer a strict M1/M2 phenotype, but rather presents a continuum along the two extremes. M1 macrophages are activated by IFNγ secreted by type 1 T-helper (Th1) cells, and by Toll-like receptor (TLR) ligands, such as LPS. They exhibit a pro-inflammatory profile and produce TNFα, IL-6 and IL-12 as well as reactive oxygen and nitrogen species, providing them with the tools needed for microbicidal actions [122]. In contrast, cytokines such as IL-4 and IL-13 that are secreted by type 2 T-helper (Th2) cells provoke M2 polarization, leading to the expression of anti-inflammatory mediators, such as TGFβ, IL-10 and IL-1 receptor antagonists, and resulting in high phagocytotic activity [122,123]. For the heart, it has been shown that in response to cardiac ischemia, the first wave of infiltrating macrophages mainly comprises M1-like macrophages, involved in initiating an inflammatory response, whereas the second wave consists of M2-like macrophages involved in tissue repair [124].

Polarization toward an M1-activated state leads to a metabolic switch, marked by an increased glycolytic flux along with an increased expression of the glucose transporter GLUT1 and various glycolytic enzymes [125,126,127,128]. Most of the glucose is converted into lactate rather than oxidized, indicative of aerobic glycolysis [129]. Notably, overexpression of GLUT1 in mouse macrophages has been shown sufficient to promote an M1-like phenotype, demonstrating the dependence of macrophage phenotype on metabolism [127].

The increased aerobic glycolysis in activated M1 macrophages also fuels the oxidative branch of the PPP, thereby promoting the formation of NADPH needed to facilitate microbicidal respiratory bursts via the NAPDH oxidase [127,130]. The mitochondrial oxidation of fatty acids is reduced in LPS-stimulated M1 macrophages [129,131]. The decline in mitochondrial oxidative phosphorylation has been linked to altered TCA cycle activity, more specifically to breaks at the level of isocitrate dehydrogenase and of succinate dehydrogenase (SDH), along with a rise in citrate and succinate levels [132,133]. Succinate stabilizes HIF-1a and thereby stimulates glycolysis and induces secretion of pro-inflammatory cytokines such as IL-1b [29,132]. Part of the citrate is transported from the mitochondria to the cytosol where ACLY can convert it into oxaloacetate and acetyl-CoA required for fatty acid synthesis. The switch to glycolytic ATP production results in mitochondrial hyperpolarization, which together with succinate-driven reverse electron transport involving respiratory complex I and II greatly stimulates mitochondrial ROS production in respiratory complex I [132].

By contrast, in IL-4-activated M2 macrophages, uptake and oxidation of fatty acids are increased due to transcriptional activation via STAT6 and PGC-1b leading to increased expression of enzymes involved in fatty acid uptake, mitochondrial transport and beta-oxidation [134]. Moreover, glutamine uptake and anaplerotic glutaminolysis are enhanced in M2 macrophages, thereby stimulating the HBP and TCA cycle activity. The glutamine-dependent generation of alpha-ketoglutarate strongly promotes the anti-inflammatory M2 phenotype [135]. The detailed metabolic studies of macrophages and other immune cells provide unequivocal evidence of the biological significance of metabolic rewiring in cells, other than tumor cells.

### 3.2. Endothelial Cell Metabolism

Healthy endothelial cells inhibit the proliferation of smooth muscle cells, adhesion of leukocytes, aggregation of platelets, and maintain appropriate vasodilation via the release of vasoactive substances [136]. Within the myocardium, cardiomyocytes are in direct interaction with endothelial cells via their adjacent position to capillaries, which ensures an adequate supply of nutrients and oxygen [137]. Although endothelial cells have immediate access to oxygen provided by the circulating blood, they do not rely on oxidative metabolism and are instead highly glycolytic [138]. Approximately 85% of ATP production in endothelial cells is achieved via aerobic glycolysis, a dependence similar to highly glycolytic cancer cells [139]. These high glycolytic rates subsequently provide glycolytic intermediates to fuel side pathways, such as the PPP and HBP, which play a role in maintaining cell viability and migration [140]. Notably, under conditions of hyperglycemia, overactivation of the HBP promotes O-linked glycosylation of signaling proteins, eventually impairing endothelial nitric oxide synthase (eNOS) activity and angiogenic capacity [141,142].

The low oxygen need of endothelial cells is advantageous, as it allows endothelial cells to sprout into low-oxygenated tissue and to spare the available oxygen for surrounding perivascular cells [143]. Although endothelial cells are capable of increasing oxidative phosphorylation when needed, they primarily produce ATP via glycolysis to generate new blood vessels [139]. The role of other metabolic pathways in endothelial cells has not been completely elucidated yet. However, endothelial cells utilize carbons derived from fatty acids for the production of nucleotide precursors as well as glutamate and aspartate for angiogenic proliferation [144,145]. Pharmacological inhibition of carnitine palmitoyl transferase-1 (CPT1), a rate-limiting enzyme of mitochondrial fatty acid oxidation, did not disturb energy balance, but inhibited nucleotide synthesis required for DNA replication and in this way reduced endothelial cell proliferation [145].

Following exposure to pro-angiogenic growth factors, endothelial cells switch to an active state and become migratory tip cells or proliferating stalk cells. Under VEGF stimulation, GLUT1 expression is increased, leading to higher glucose transport in proliferating endothelial cells [146]. Nearly all glucose is converted into lactate and excreted, and thus not used for the synthesis of nucleic acids or amino acids. Accordingly, in contrast to cancer cells, the high glycolytic activity of endothelial cells is not meant to sustain anabolic processes in the proliferating cells. However, glutamine remains within endothelial cells, indicating a significant contribution of this amino acid in the generation of the biomass needed for cell proliferation. Pharmacological or genetic inhibition of glutamine utilization leads to a marked reduction in TCA cycle intermediates and induces proliferative arrest [147,148]. Due to their dependence on glycolysis, endothelial cells contain fewer mitochondria than cells driven by oxidative phosphorylation, and experimental inhibition of mitochondrial ATP production does not affect vessel sprouting [139]. By contrast, suppression of glycolysis in endothelial cells by inhibiting phosphofructokinase-2/fructose-2,6-biphosphatase 3 (PFKFB3) or PKM2, potently inhibits angiogenesis by inhibiting endothelial cell migration and proliferation [149,150,151]. The latter observations support the concept that therapeutic interventions at the level of metabolism can be used to modulate angiogenesis [152].

### 3.3. Vascular Smooth Muscle Cell Metabolism

An important function of vascular smooth muscle cells (VSMCs) is to regulate vessel tone, and thereby blood flow and blood pressure. Due to their specialized role, VSMCs express a distinctive set of signaling molecules and contractile proteins, including smooth muscle alpha-actin, smooth muscle myosin heavy chain and calponin [153]. Within mature, healthy blood vessels, they maintain a low proliferative rate and produce small amounts of ECM proteins [153]. However, in response to mechanical or neurohumoral stimulation secondary to hypertension or vascular injury, VSMCs undergo phenotypic switching toward a more synthetic phenotype. Synthetic VSMCs are characterized by increased cellular proliferation and migration in addition to the upregulated synthesis of ECM components and decreased expression of contractile proteins [153].

VSMCs exhibit a Warburg-like metabolism, which is further increased after acquiring a synthetic phenotype. VSMCs utilize aerobic glycolysis to source a third of the total ATP, and 90% of this glycolytic flux results in the generation of lactate [154]. Increased levels of lactate in the extracellular environment further reinforce the VSMC synthetic phenotype [155]. Oxidative phosphorylation is also used to produce ATP. If fatty acid oxidation is specifically stimulated, aerobic glycolysis is partially substituted, indicating that VSMCs exhibit metabolic flexibility and that aerobic glycolysis is not essential in all metabolic conditions [156]. Exposure of VSMCs to 2-deoxyglucose, a glucose analog inhibiting glycolysis at the level of HK, reduces proliferation in a dose-dependent fashion, showing that aerobic glycolysis, although not essential for cell survival and maintenance of the cytoskeleton, nevertheless plays an important role in VSMCs proliferation [157]. High glucose environments as well as induced upregulation of cellular glucose metabolism promote anti-apoptotic pathways in VSMCs via protein kinase C (PKC) activation [158] and inactivation of glycogen synthase kinase 3-beta (GSK3b) [159]. The increase in glycolytic activity in synthetic VSMCs partially depends on elevated HK2 expression secondary to the activation of STAT3 or the PI3K-AKT pathway [160,161]. Pharmacological inhibition of HK2 prevents the migration as well as the proliferation of pulmonary VSMCs. Likewise, activation of ENO1 stimulated the proliferation and de-differentiation of pulmonary VSMCs [162], once more illustrating that metabolic cues drive the transition from a contractile to synthetic VSMC phenotype.

### 3.4. Fibroblast Metabolism

Fibroblasts provide structural support to the myocardium via the synthesis of extracellular matrix (ECM) components [163]. Under physiological conditions, ECM homeostasis is maintained via cell–matrix interactions and biochemical signaling in so-called quiescent fibroblasts. However, pathological stimuli cause a switch from a quiescent to an activated state, eventually resulting in their transition into myofibroblasts. Myofibroblasts are characterized by elevated levels of alpha-smooth muscle actin (aSMA) containing stress fibers and display increased collagen production, mobility and contraction [164]. TGF-beta, a cytokine commonly produced by M2 macrophages in the process of wound repair, is a potent inducer of fibroblast to myofibroblast differentiation [165].

Fibroblast activation is both induced and accompanied by metabolic reprogramming toward a more glycolytic state: High glucose environments activate cardiac and kidney fibroblasts and promote their proliferation and transition into myofibroblasts in vitro [166,167]. Likewise, cardiac fibroblasts isolated from diabetic rat hearts exhibit a myofibroblast phenotype characterized by increased aSMA expression and contractility [168]. Fibroblasts from the lung tissue of pulmonary hypertension patients or fibrotic kidneys exhibit increased glycolytic gene expression as well, despite deriving from a normoglycemic environment [166,169]. Induction of myofibroblasts via TGF-beta leads to an increased expression of glycolytic enzymes such as HK2, PFK and LDHA via HIF-1a [166,170,171]. The inhibition of glycolysis with PFKFB3 inhibitors in turn reduced TGF-beta-induced effects in vitro and attenuated the development of fibrosis in a murine model of lung fibrosis [172]. The increased aerobic glycolysis leads to an accumulation of succinate, stabilizing HIF-1a and further stimulating fibroblast activation [172,173]. Likewise, hypoxia enhanced myofibroblast formation via HIF-1a in a mouse model of lung fibrosis [174]. Enhanced mitochondrial fragmentation through the overexpression of mitochondrial fission factor (MFF) limits mitochondrial respiration and enforces a glycolytic shift, thereby also inducing myofibroblast-like characteristics such as increased aSMA and calponin expression [175]. On the other hand, supplementation with 2-deoxyglucose, which effectively inhibits aerobic glycolysis, leads to decreased aSMA expression and a less pronounced myofibroblast phenotype [171]. The metabolic shift toward increased glycolysis in cardiac fibroblasts is accompanied by transcriptional activation and secretion of ECM proteins as well as increased O-GlcNAcylation via increased glucose flux through the HBP [176,177]. Apart from glycolytic reprogramming, myofibroblasts also exhibit increased glutaminolysis, providing the cells with citric acid cycle intermediates necessary for their increased energetic demands [173]. Inhibition of glutaminolysis via reduced glutaminase activity in mice with liver fibrosis suppressed myofibroblast accumulation and the progression of fibrosis [173]. The myofibroblast phenotype can also be reversed, as has been documented for hepatic stellate cell-derived myofibroblasts: These cells revert to a quiescent state when HIF-1a, glycolytic enzymes, glutaminolysis or lactate accumulation are inhibited [178]. Zhao and colleagues reported that the inhibition of glycolysis as well as the stimulation of fatty acid oxidation reduced the production of ECM proteins by dermal fibroblasts [179]. Collectively, these observations lend support to the concept that cardiac fibrosis may also be reversed via metabolic interventions.

### 3.5. Cardiomyocyte Metabolism

As reviewed in the previous paragraphs, phenotypical switches in the various cell types that constitute the cardiac muscle are associated with changes in cellular metabolism and, conversely, forced changes in cellular metabolism influence cellular phenotype and function. In view of the interest in the significance of “energy starvation” as a causative factor in the development of cardiac disease, it is quite surprising that little is known about the significance of metabolic rewiring in cardiomyocyte remodeling. To some extent, this can be explained by the fact that adult cardiomyocytes are non-proliferating, terminally differentiated cells that are notoriously difficult to study under in vitro conditions. Some information can be extracted from in situ studies that made use of cardiomyocyte-restricted knockout or overexpression of genes involved in fatty acid or glucose metabolism. Table 2 provides a summary of such studies, focusing on the effect of specific genetic interventions on intermediary metabolism and on the development of cardiac hypertrophy. It is worth noting that every intervention which limits fatty acid metabolism, irrespective of whether it is at the level of cardiomyocyte fatty acid uptake or mitochondrial beta-oxidation, led to an increase in glucose metabolism and was associated with increased hypertrophy. Conversely, stimulation of fatty acid metabolism by the cardiac-restricted knockout of acetyl-CoA carboxylase (ACC) had the opposite effect. These observations fit with the idea that increased glycolysis is linked to hypertrophic growth. However, the outcomes of studies with interventions at the level of glucose metabolism are more divergent (Table 2), being influenced by the level at which glycolysis is blocked. In some cases, glycolytic flux was affected in its entirety. In other cases, such as the cardiac-restricted overexpression of a kinase-deficient PFK2, the PPP and HBP were stimulated, while at the same time glycolytic flux was reduced [180,181].

Alternatively, compensation by other isoforms of the inhibited enzyme, or differences in the measurement of glycolytic flux might be involved. To illustrate, cardiac-restricted knockout of GLUT4 led to a counterintuitive increase rather than a decrease in glycolysis, due to a compensatory increase in cardiac GLUT1 expression [191].

Although adult cardiomyocytes are considered to be terminally differentiated, it has also been demonstrated that hypertrophy leads to the partial re-activation of the cell cycle and the re-expression of fetal genes [20,195]. The neonatal heart is highly dependent on glucose as substrate and neonatal cardiomyocytes still have the ability to proliferate (reviewed in [196]). During cardiac postnatal development, the loss of the ability to proliferate coincides with the switch from glycolysis to mitochondrial fatty acid oxidation. Along this line of thinking, attempts are currently being made to intervene in the metabolism of induced pluripotent stem cell (iPSC)-derived cardiomyocytes, with the aim of promoting a more differentiated, oxidative, adult-like phenotype of these cells [197].

In hypertrophying cardiomyocytes, the re-entry of the cell cycle leads to polyploidy and binucleation instead of cell division and allows cardiomyocytes to increase in size. It is tempting to speculate that, in analogy to proliferating cancer cells, this incomplete cytokinesis represents the attempt of hypertrophic cardiomyocytes to initiate proliferation without the need to give up their structural integrity and function. Following this line of reasoning, the hypertrophic switch to a more glycolytic metabolism and the re-expression of fetal genes should be considered part of the proliferative growth program.

## 4. Metabolic Cross-Talk between Cells

It is common knowledge that at the whole-body level there is metabolic interdependency between tissues. Emerging evidence indicates that there is also extensive metabolic cross-talk between the different cell types within an organ. For instance, within skeletal muscle, lactate is exchanged between white and red muscle fibers [198]. Equally intriguing is the observation that lactate produced by tumor cells via the process of aerobic glycolysis induces the polarization of tumor-associated macrophages toward an M2-like phenotype [199]. It has also been proposed that the secretion of lactate by tumor cells has a damaging effect on non-cancerous neighboring cells through the acidification of the microenvironment [200,201].

Research on metabolite signaling has been boosted by the recent discovery of G-protein coupled receptors that act as selective metabolite sensors, for instance for long-chain fatty acids (GPR40, GPR120) [202], succinate (GPR91) [203] and lactate (GPR81) [204]. In fact, lactate binding to GPR81 interferes with Toll-like receptor-mediated activation of the inflammasome (NLPR3), thereby providing a mechanistic explanation for the earlier noted anti-inflammatory effect of lactate [205]. Lactate has also been found to inhibit monocyte migration and the secretion of proinflammatory cytokines [206]. However, it is not always easy to discern if observed effects are related to lactate–GPR81 signaling, to metabolic effects (substrate competition), or to lactate-mediated acidification of the microenvironment, all of which elicit cellular responses. Notably, pH lowering appears to promote exosome secretion and, in this way, cross-talk between cells [207].

Succinate accumulates in the myocardium under certain pathological conditions, particularly low oxygen states, and is subsequently released into the circulation [208,209]. Succinate-mediated activation of GPR91 has been found to induce renovascular hypertension [203] and cardiomyocyte hypertrophy in vitro and in vivo [210]. These observations suggest that the myocardium also makes use of metabolic signals to communicate between (neighboring) cell types. It will be interesting to explore in what manner the activation (or inhibition) of these recently discovered metabolic receptors modulates cardiac phenotype and function.

## 5. Metabolic Rewiring in the Failing Heart

It is commonly acknowledged that cardiac hypertrophy and failure are associated with alterations in cardiac substrate and energy metabolism [5,7,11]. However, even after decades of research, it is still debated if this type of metabolic remodeling should be considered an adaptive or maladaptive response [211].

The picture even becomes more complicated when the distinct forms of heart failure are taken into consideration, namely heart failure with reduced ejection fraction (HFrEF) and with preserved ejection fraction (HFpEF). HFrEF is characterized by systolic dysfunction and is associated with reduced fatty acid and increased cardiac glucose uptake. As such, the switch in cardiac metabolism in HFrEF fits the traditional picture. Generally speaking, the switch from fatty acids to glucose has usually been considered an energy-efficient, adaptive response, given that the oxidation of glucose yields more ATP per oxygen atom used, than that of fatty acids (reflected in a higher ATP/O ratio for glucose). However, it is well-established that in the hypertrophic and failing heart a substantial part of the glucose is not oxidized, but converted into lactate instead [212], reducing the total ATP yield [34,35]. Whether the lactate is produced via anaerobic or aerobic glycolysis remains to be determined. If aerobic glycolysis (Warburg effect) is involved, this would imply that a mismatch between oxygen supply and demand is not necessarily the driving force for the increase in cardiac lactate production. As discussed in this review, it is well conceivable that the increase in glycolytic flux primarily serves to support cardiac anabolic processes instead of sustaining energy metabolism.

By contrast, HFpEF is characterized by diastolic dysfunction. In Western societies, the prevalence of HFpEF is rapidly increasing and is more frequently observed in individuals with obesity and type 2 diabetes [213]. These individuals usually suffer from insulin resistance, which profoundly affects systemic metabolism. Under these conditions, the heart is forced to rely even more on the use of fatty acids as substrate. If we assume that glycolytic metabolism is required to support hypertrophic growth, then the inability to increase glycolysis should have consequences for the structural remodeling of the heart. In line with this is the observation that in a pig model of cardiometabolic disease, cardiomyocyte size was reduced compared to the controls, despite the presence of marked hypertension [214]. Furthermore, earlier studies by Stanley et al. (reviewed in [215]) showed that a high-fat diet attenuated the increase in cardiac mass in rats subjected to pressure overload. These findings support the idea that the hypertrophic growth of cardiomyocytes depends on an increased glycolytic flux.

This line of thinking will also have consequences for the design of metabolic therapies for the diseased heart. There is an increasing interest in the use of metabolic interventions, as an adjunct to current pharmacological therapies that target the various neuro-hormonal pathways, to treat heart failure [2]. Generally speaking, the goal of metabolic therapies has been to improve cardiomyocyte energy metabolism by stimulating ATP synthesis. Primarily because of the higher ATP/O ratio of glucose, so far nearly all metabolic interventions in clinical studies with HF patients have been aimed at promoting cardiomyocyte glucose utilization through pharmacological inhibition of enzymes involved in mitochondrial fatty acid uptake and beta-oxidation [2]. However, what effect will this have on other cell types? For instance, is it possible that by inhibiting fatty acid oxidation and by stimulating glycolysis one promotes cardiac inflammation by favoring the formation of the more pro-inflammatory M1 macrophages over the anti-inflammatory M2 macrophages? The limited success of metabolic interventions so far may well be related to the fact that these interventions not only increase glycolytic flux and alter cardiomyocyte energy metabolism, but they also modulate the phenotype and function of other cardiac cell types.

Surprisingly, one of the most promising classes of drugs now appear to be the sodium-glucose co-transporter-2 inhibitors (SGLT2is), glucose-lowering anti-diabetic drugs that directly interfere with renal glucose reabsorption. Despite the absence of the sodium-glucose co-transporter-2 in the heart, there is accumulating clinical evidence that SGLT2is have marked beneficial effects in the setting of both HFrEF and HFpEF. It still remains to be determined if the beneficial outcome is related to their effects on systemic metabolism, cardiomyocyte metabolism or that of other cell types in the heart, due to off-target effects, or to a combination of those [216].

## 6. Concluding Remarks and Future Perspectives

So far, changes in cardiac metabolism in the failing heart have been primarily considered from the perspective of the consequences for the amount of ATP needed to maintain cardiac pump function. As discussed above, there is accumulating evidence that, similar to cancer cells, the different cell types constituting the cardiac muscle make use of metabolic rewiring to adjust their phenotype and function. Consequently, in the context of the diseased heart, metabolic rewiring provides a mechanism to adjust cellular phenotype and function and, as such, is likely to play a role in many, if not all, aspects of cardiac structural remodeling, that is cardiac hypertrophy, fibrosis, angiogenesis and inflammation.

Unfortunately, current research on cardiac metabolism does not allow one to discriminate between different cell types in the heart. When using tracers to monitor metabolism in vivo or ex vivo using isolated heart preparations, the cardiac muscle is studied as one entity and it is assumed that the measurements reflect the metabolism of contracting cardiomyocytes. For a large part, this assumption is correct given that cardiomyocytes constitute the largest volume fraction of the myocardium and are the main energy-consuming units of the cardiac muscle. Nonetheless, it is equally important to assess how the other cell types in cardiac muscle tissue adjust their metabolism during the development of cardiac hypertrophy and failure, and how these cells respond to metabolic therapies. Single-cell RNA-sequencing analyses of the different cell populations in the healthy and diseased heart are already providing useful information about the metabolic setup of the different cell types under these conditions. Additionally, in vitro strategies such as cellular metabolic flux analysis and live cell imaging will assist in unraveling the exact nature of the relationship between metabolic rewiring and cellular phenotype. In the near future, mass spectrometry imaging might provide a useful tool to look at, quite literally, the metabolism of different cell types in the heart in situ, especially since technical developments already allow a resolution in the micrometer range of the acquired metabolic images [217]. Undoubtedly, the implementation of such novel techniques will deepen our understanding of the significance of metabolic rewiring in cardiac disease.

In summary, in the present review we provide an alternative point of view on the significance of metabolic remodeling in the diseased heart, diverging somewhat from the traditional concept that adjustments in cardiac metabolism merely serve to secure ATP production as much as possible and to maintain cardiac pump function at an adequate level. Specific points that we would like to present as “substrate for thought” are: Heart failure pathogenesis is associated with marked alterations in cardiac substrate and energy metabolism, the relevance of which is still incompletely understood.Similar to cancer cells, metabolic rewiring not only regulates energy balance but also affects the phenotype and function of the principal cell types of the cardiovascular system.Cardiac lactate production does not necessarily reflect a shortage in oxygen supply. It can also be due to the Warburg effect.Glycolytic enzymes have multiple biological functions, exceeding their role as catalyzers of chemical reactions in the glycolytic pathway.Metabolic signaling is a still underdeveloped area of research that is likely to have marked implications for our understanding of cardiac pathogenesisThe switch toward enhanced glycolysis in cardiac hypertrophy and the polyploidization and binucleation of cardiomyocytes may be related events.The various cell types in the heart have different substrate demands and respond differently to disease- or pharmacologically-induced changes in metabolism. This should be taken into account when designing metabolic therapies aimed at improving cardiac function.

To conclude, particularly developments in research areas outside the cardiovascular arena force us to take a look at disease-associated changes in cardiac substrate metabolism from other angles. After decades of research, perhaps this change in paradigm may eventually lead to a breakthrough in elucidating if metabolic remodeling is adaptive or maladaptive and in deciding which metabolic therapies may have long-term favorable effects on the failing heart.

## Figures and Tables

**Figure 1 ijms-23-13902-f001:**
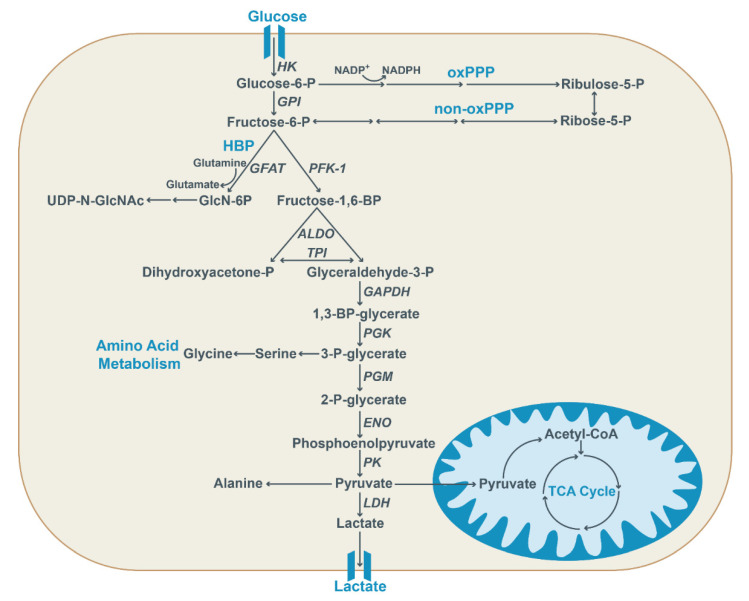
Glycolysis provides important intermediates for anabolic processes. Glucose is transported into the cell via glucose transporters and first converted into glucose-6-phosphate by hexokinase (HK) and subsequently converted into fructose-6-phosphate. Alternatively, it acts as a substrate for the oxidative pentose phosphate pathway (oxPPP) to generate ribulose-5-phosphate, thereby also generating reducing equivalents in the form of NADPH. Fructose-6-phosphate is converted into fructose-1,6-phosphate by phosphofructokinase 1 (PFK-1) or it can be utilized in the non-oxidative pentose phosphate pathway (non-oxPPP), ultimately resulting in ribose-5-phosphate generation. Moreover, the hexosamine biosynthetic pathway (HBP) utilizes fructose-6-phosphate, together with glutamine, to generate glucosamine-6-phosphate (GlcN-6P) via glutamine:fructose-6-phosphate aminotransferase (GFAT). Subsequently, the end product UDP-N-acetylglucosamine (UDP-N-GlcNAc) is generated in a multi-step conversion. Glyceraldehyde-3-phosphate generated by aldolase (ALDO) and triose-phosphate isomerase (TPI) further undergoes glycolysis. After conversion into 1,3-BP-glycerate via glyceraldehyde-3-phosphate-dehydrogenase (GAPDH), the glycolytic intermediate 3-P-glycerate is generated via phosphoglycerate kinase (PGK) and converted into pyruvate by pyruvate kinase (PK). It can also be used for the synthesis of amino acids. Pyruvate is either converted into lactate by lactate dehydrogenase (LDH), used to generate amino acids, or enters the mitochondria to fuel the TCA cycle.

**Figure 2 ijms-23-13902-f002:**
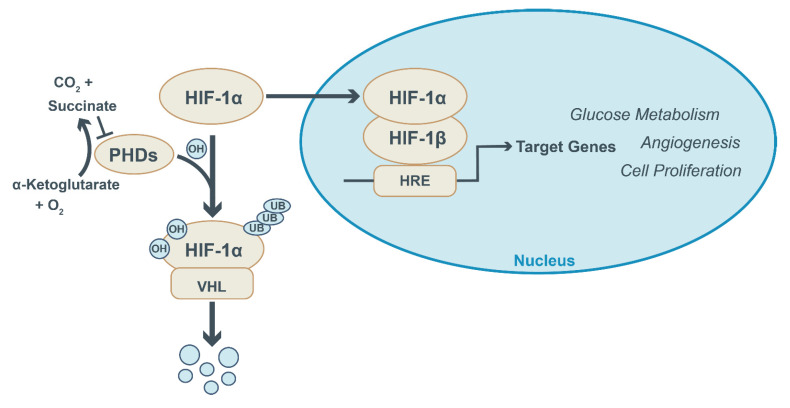
HIF-1a signaling and regulation. Under normoxic conditions, prolyl hydroxylase enzymes (PHDs) hydroxylate HIF-1a and mark it for degradation by the von Hippel–Lindau E3 ubiquitin ligase (VHL). HIF-1a hydroxylation by PHDs requires alpha-ketoglutarate and oxygen as substrates and yields succinate and CO_2_. PHD activity is suppressed under hypoxic conditions and following the accumulation of succinate. This prevents HIF-1a degradation and allows its translocation into the nucleus, where it dimerizes with HIF-1b and binds to HIF-responsive elements (HRE) in the promotor region of genes. This activates the expression of target genes involved in glucose metabolism, angiogenesis and cellular proliferation.

**Figure 3 ijms-23-13902-f003:**
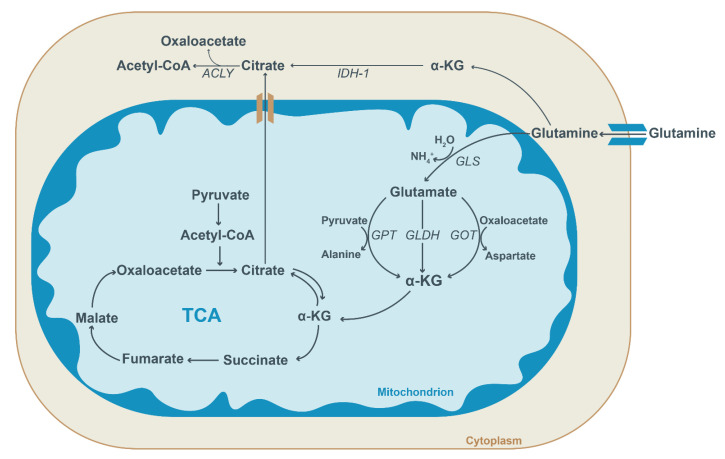
Schematic overview of glutamine metabolism. In the cytosol, glutamine is converted into alpha-ketoglutarate (α-KG) or transported into mitochondria. Within the mitochondria, glutamine is first converted into glutamate by glutaminase (GLS) and then to α-KG. Glutamate conversion into α-KG is facilitated by three enzymes: glutamate pyruvate transaminase (GPT), glutamate dehydrogenase (GLDH) and glutamate oxaloacetate transaminase (GOT). Subsequently, α-KG enters the TCA cycle and aids the generation of succinate, thereby replenishing the TCA cycle (anaplerosis). Alternatively, it can be used to generate citrate (reductive carboxylation) which can be transported to the cytosol. In the cytoplasm, isocitrate dehydrogenase (IDH-1) facilitates the conversion of α-KG into citrate. Citrate is utilized for acetyl-CoA and oxaloacetate generation by ATP-citrate lyase (ACLY). Cytosolic acetyl-CoA forms the substrate for lipogenesis and the acetylation of proteins.

**Table 2 ijms-23-13902-t002:** Cardiac-restricted deletion or overexpression of metabolic genes affects hypertrophic growth.

Gene	Genetic Intervention	Metabolic Consequences	Effect on Hypertrophy	Ref.
FA Metabolism				
CD36	KO	FAO↓ GO↑ Glc↑	↑	[182]
ACSL	KO	FAO↓ GO↑	↑	[183,184]
CPT1b	KO	n.d.	↑	[185]
CPT2	KO	FAO↓	↑	[186]
ACC2	KO	FAO↑ GO↓	↓	[187,188]
Glucose Metabolism				
GLUT1	KO	FAO↑ GO↓ Glc↓	=	[189]
GLUT1	OE	GO↑ Glc↑	↑	[190]
GLUT4	KO	Glc↑	↑	[191]
HK2	OE	FAO↓ GO↑	↓	[84]
PFK2	OE	Glc↓	↑	[180]
MPC1	KO	FAO↑ GO↓ Glc=	↑	[192,193]
PDH-a1	KO	FAO = GO↓ Glc=	↑	[194]
PDK4	OE	FAO↑ Glc↓	=	[194]

Reported effects of cardiomyocyte-restricted knockout (KO) or overexpression (OE) on cardiac metabolism and hypertrophy. FAO, GO and Glc refer to fatty acid oxidation, glucose oxidation and glycolysis, respectively. Symbols refer to the effect on metabolism and cardiac hypertrophy (enhanced ↑, reduced ↓, unchanged =). ACSL—acyl-CoA synthase; MPC—mitochondrial pyruvate carrier; n.d.—not determined.

## Data Availability

Not applicable.

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
