# Peer review of "Oncometabolism: A Paradigm for the Metabolic Remodeling of the Failing Heart"

_ijms, 2022, doi:10.3390/ijms232213902_

Round 1

Reviewer 1 Report

In this review, the authors have compared metabolism-driven phenotypical alterations in cardiac cells and their potential relevance in the pathophysiology of cardiac disease.

The review is timely and overall well‐reasoned, and well-written. The authors have complemented this review by including excellent schematics.

I am satisfied with scientific content and presentation of the subject. I have no further comments.

Author Response

We would like to thank the reviewer for the highly positive assessment of our manuscript both in terms of scientific content and format presentation.

Reviewer 2 Report

Dear Authors, The review entitled Oncometabolism: A paradigm for the metabolic remodeling of the failing heart is a comprehensive analysis of the data published in recent years on cancer and cardiac metabolism, focusing mainly on its regulation and adaptation to pathological conditions. It is a very interesting collection of information, divided into important main chapters. However, despite the authors' great efforts, the manuscript is very difficult to understand because of the huge collection of information that is not discussed in the epilog. In fact, the data are presented without drawing any conclusion or expressing any point of view on the results presented. I suggest that you write a concluding comment in each of your paragraphs to clarify the key message of the manuscript. In addition, the manuscript should include more suggestions for further research/experiments that should be performed to characterize metabolic rewiring in cardiac tissue/cells. I also suggest adding a final figure summarizing the authors' conclusions.
Minor point: check typos and double spaces (eg, Lane 54, 80, 99, 135, 178, 209, 294, 304, 334, 451, 524, 527, 594, 603, 627)

Author Response

The review …. is a comprehensive analysis of the data published in recent years on cancer and cardiac metabolism, focusing mainly on its regulation and adaptation to pathological conditions.

It is a very interesting collection of information, divided into important main chapters. However, despite the authors' great efforts, the manuscript is very difficult to understand because of the huge collection of information that is not discussed in the epilog. In fact, the data are presented without drawing any conclusion or expressing any point of view on the results presented. I suggest that you write a concluding comment in each of your paragraphs to clarify the key message of the manuscript.

In addition, the manuscript should include more suggestions for further research/experiments that should be performed to characterize metabolic rewiring in cardiac tissue/cells. I also suggest adding a final figure summarizing the authors' conclusions.

We would like to thank the reviewer for the helpful comments. Along the line of the reviewer’s suggestions adjustments have been made to the revised manuscript to increase readability. The following changes were implemented:

  • In the revised version of the manuscript, we used headings and subheadings to better organize the paragraphs by topic.
  • As suggested by the reviewer at various places in the text, at the start of a new paragraph or at the end of a paragraph, sentences were added to provide more context to the text (under "track changes" in the revised manuscript).
  • We largely restructured the final paragraph (“Concluding remarks and future perspectives”) to make it more coherent with the preceding text and to avoid introducing new topics. Therefore, (1) some parts were removed entirely (the part about cellular survival), (2) the part about metabolic therapy was moved to the previous paragraph (also because reviewer 3 suggested to pay attention to SGLT2 inhibitors as well) and importantly (3) alongside additional suggestions for further research, we added specific “take home messages” that summarize the main points that we try to convey in the preceding paragraphs. We consider the addition of the take home messages as a useful addition and a good alternative for a summarizing figure.

Minor point: check typos and double spaces (eg, Lane 54, 80, 99, 135, 178, 209, 294, 304, 334, 451, 524, 527, 594, 603, 627)

The text was checked. Typos, double spaces, etc., have been removed. We also noticed that upon submission of the original manuscripts all subscripts and superscripts (reference numbers) were removed. This has now been corrected (references are now set in between square brackets).

Reviewer 3 Report

Overall this review is well written and comprehensively covers the process of metabolic rewiring that occurs in the failing heart.   There only a few minor aspects to address.

1.  The reference formatting was distracting as written and should be properly formatted.

2. The authors mention that fibroblasts transdifferentiate.  This process is more accurately described now as fibroblast activation.

3. The authors could consider discuss the recent clinical successes of SGLT2 inhibitors in heart failure trials with respect to potential actions on metabolism.

4. The authors state "fuel important side branches of the glycolytic pathway such as the PPP"    I think this should be rewritten as "glucose is diverted (or shunted) into the PPP."  As the PP is separate from the glycolysis pathway.

Author Response

We would like to thank the reviewer for the positive assessment of the manuscript and the helpful comments. Along the line of the reviewer’s suggestions adjustments have been made to the revised manuscript. Please, find below an itemized response to the specific comments.

  1. The reference formatting was distracting as written and should be properly formatted.

Apparently, upon submission of the original manuscript all subscripts and superscripts (reference numbers) had been removed. This has now been corrected (references are now set in between square brackets).

  1. The authors mention that fibroblasts transdifferentiate. This process is more accurately described now as fibroblast activation.

As suggested by the reviewer the paragraph on fibroblast metabolism (paragraph 3.4 in the revised version, page 13) was adjusted at various points, now using the more timely terminology “fibroblast activation” instead of “transdifferentiation” to describe the switch from fibroblast to myofibroblast.

  1. The authors could consider discuss the recent clinical successes of SGLT2 inhibitors in heart failure trials with respect to potential actions on metabolism.

In the revised version a couple of sentences (paragraph 5, page 16-17))  were devoted to the - so far incompletely understood - cardiac benefit of these drugs. At the same time, to avoid addressing the topic metabolic therapy at multiple points in the text, we moved the short section on metabolic therapy from the final paragraph to paragraph 5.

  1. The authors state "fuel important side branches of the glycolytic pathway such as the PPP" . I think this should be rewritten as "glucose is diverted (or shunted) into the PPP."  As the PP is separate from the glycolysis pathway.

We agree with the reviewer that our description about the relation of PPP to glycolysis was incorrect. In the revised version we now open the paragraph on the pentose phosphate pathway (paragraph 2.3, page 7, in the revised version) with the sentence “As discussed, the increased uptake and conversion of glucose to glucose-6-phosphate also fuels the PPP.”  A sentence in paragraph 2.1 (page 3, line 109) was adjusted along the same line.

Round 2

Reviewer 2 Report

Dear authors, I really appreciate the effort to make the manuscript more clear and organized. I suggest to check English style of the new part you add.

Thank you very much and best regards.